# Antiparasitic Properties of Cantharidin and the Blister Beetle *Berberomeloe majalis* (Coleoptera: Meloidae)

**DOI:** 10.3390/toxins11040234

**Published:** 2019-04-22

**Authors:** Douglas W. Whitman, Maria Fe Andrés, Rafael A. Martínez-Díaz, Alexandra Ibáñez-Escribano, A. Sonia Olmeda, Azucena González-Coloma

**Affiliations:** 1School of Biological Sciences, Illinois State University, Normal, IL 61790, USA; dwwhitm@ilstu.edu; 2Instituto de Ciencias Agrarias, CSIC, Serrano 115-dpdo, 28006 Madrid, Spain; mafay@ica.csic.es; 3Facultad de Medicina, Universidad Autónoma de Madrid (UAM), Arzobispo Morcillo S/N, 28029 Madrid, Spain; rafael.martinez@uam.es; 4Facultad de Farmacia, Universidad Complutense de Madrid (UCM), CEI Campus Moncloa, 28040 Madrid, Spain; alexandraibanez@ucm.es; 5Facultad de Veterinaria, Universidad Complutense (UCM), 28040 Madrid, Spain; angeles@ucm.es

**Keywords:** cantharidin, blister beetle, *Berberomeloe majalis*, nematicide, ixodicide, antifeedant

## Abstract

Cantharidin (CTD) is a toxic monoterpene produced by blister beetles (Fam. Meloidae) as a chemical defense against predators. Although CTD is highly poisonous to many predator species, some have evolved the ability to feed on poisonous Meloidae, or otherwise beneficially use blister beetles. Great Bustards, *Otis tarda*, eat CTD-containing *Berberomeloe majalis* blister beetles, and it has been hypothesized that beetle consumption by these birds reduces parasite load (a case of self-medication). We examined this hypothesis by testing diverse organisms against CTD and extracts of *B. majalis* hemolymph and bodies. Our results show that all three preparations (CTD and extracts of *B. majalis*) were toxic to a protozoan (*Trichomonas vaginalis*), a nematode (*Meloidogyne javanica*), two insects (*Myzus persicae* and *Rhopalosiphum padi*) and a tick (*Hyalomma lusitanicum*). This not only supports the anti-parasitic hypothesis for beetle consumption, but suggests potential new roles for CTD, under certain conditions.

## 1. Introduction

Cantharidin (CTD) is a toxic trycyclic monoterpene with the chemical formula: 3,6-epoxy-1,2-dimethylcyclohexane-1,2-dicarboxylic anhydride (Figure 1). Found in blister beetles, CTD was one of the first pharmacoactive natural products used by humans [1,2,3], and was long considered a sexual stimulant [4,5,6,7,8]. In the late Middle Ages, *Lytta vesicatoria* blister beetles were collected and sold throughout Europe as an aphrodisiac, known as “Spanish Fly” (Figure 2) [9,10,11,12,13]. Today, CTD is used on humans to treat both common and molluscum warts, to remove tattoos, and as a counterirritant, and, until recently, was used as a sexual stimulant in livestock breeding [4,14]. Against vertebrates, CTD is a powerful vesicant and highly toxic. However, in low doses it “stimulates” or irritates vertebrate mucus membranes [10,15,16]. Human ingestion can result in vomiting, diarrhea, bleeding from the gastrointestinal tract, nephritis, hematuria, proteinuria, liver, kidney and other organ edema and failure, and death [4,16,17,18,19,20,21]. The consumption of beetles in fresh forage or hay, or drinking beetle- contaminated water, can seriously harm pets, poultry, or livestock [16,18,22,23].

Biochemically, CTD acts at multiple levels [16]. It is a potent and specific inhibitor of protein phosphatases 1 (PP1) and 2A (PP2A) [24,25]. It causes the release of serine proteases, which break the peptide bonds in proteins, destroying the adhesion between cells, releasing fluids and causing blistering and bleeding [4]. It disrupts mitosis [16].

CTD was first discovered in blister beetles (Order: Coleoptera; Family: Meloidae), a group of ~3000 species found in temperate and tropical regions world-wide [16,26]. Most meloids are chemically protected from predators by the presence of CTD, which also plays a role in mating [27]. CTD is transferred from males to females during mating in CTD producing insects [28]. Furthermore, CTD synthesis takes place in the male body and is finally deposited in the testes—hemolymph transport is not involved. In females, CTD enters the genitalia from the male as a nuptial gift [28]. 

A few insect predators have evolved partial immunity to CTD and, in some cases, actually use this poisonous substance for their own benefit. Some insects [27], frogs, toads [29], birds [30], and mammals [31] consume them in the wild. Other uses described include the protection of white breasted nuthatches nestholes by sweeping the bark with a meloid insect [32] or traditional pharmacological use by humans [33]. 

For example, great bustards, *Otis tarda*, a vulnerable and protected bird species in Europe, consume red-striped oil beetles, *Berberomeloe majalis*, a common CTD-containing blister beetle in the Mediterranean area, even though the beetle is highly toxic [17,34,35]. Bravo et al. (2014) [36] suggest that beetle consumption by bustards (particularly males) represents CTD self-medication to reduce parasites and diarrhea that impair the appearance of the cloaca of the birds (a central element of courtship), thus increasing their chances of reproduction.

Bravo et al.’s hypothesis is reasonable, considering that CTD is bactericidal [36], and that birds are greatly harmed by a diverse range of pathogens and parasites, including numerous bacteria, protozoa, helminths and arthropods. The protozoans *Eimeria spp*., *Cryptosporidium spp*., *Giardia spp.*, *Trichomonas spp*., *Histomonas spp*. and *Hexamita spp*. commonly infect bird digestive tracts [37]. Two protozoa cause oropharyngeal diseases in bustards: *Trichomonas gallinae* and *Entamoeba anatis* [38]. Cestodes (*Hispaniolepis sp., Raillietina cesticillus, Schistometra (Otiditaenia) conoides*, and *Idiogenes otidis*), nematodes (*Capillaria sp., Syngamus trachea, Cyathostoma sp., Heterakis gallinae, H. isolench, Aprocta orbitalis, Oxyspirura hispanica*, and *Trichostrongylus sp*), insects (including mallophaga such as *Otilipeurus turmalis*, and fly maggots such as *Lucilia sericata*) and ticks (*Rhipicephalus sanguineus*, and *Hyalomma sp.*) also infest bustards [37,39,40].

In this paper, we examine Bravo et al.’s (2014) hypothesis [36], by testing the antiparasitic efficacy of both pure CTD and extracts of *B. majalis* beetles against protozoa (*Trichomonas vaginalis*), a nematode (*Meloidogyne javanica*), and a tick (*Hyalomma lusitanicum*). Additionally, several phytophagous insects (*Myzus persicae, Rhopalosiphum padi*, *Spodoptera littoralis*) have been tested to include target species other than meloid predators or bird parasites. Our results show strong anti-parasite activity, supporting Bravo et al.’s hypothesis, and suggesting new roles for CTD.

## 2. Results and Discussion

Cantharidin (CTD) concentrations vary greatly between and within Meloid species. Various studies have found from <0.04 to 30.3 mg CTD/individual beetles [16,41,42,43]. Variation in arthropod defense titers is well known [44]. In *Berberomeloe majalis* the reported CTD content in adults varied between 0.035–1.89 mg/beetle (0.015–0.845 mg/g) [34] and 1.05–109.2 mg / beetle (1.5–156.7 mg/g) [36]. Our analysis (Table 1) indicated CTD concentrations of 295 and 41.2 µg/mg in our hemolymph extract and body extract, respectively, indicating that the hemolymph extract was ~7 times more concentrated in CTD than the body extract. The total CTD contained in 200 insects was 1819 mg (hemolymph + body), giving an estimated value of 9.1 mg of CTD per beetle. These concentrations are within the ranges reported by Bravo et al. (2014) [36]. We detected relatively low amounts of CTD in *B. majalis* hemolymph as opposed to the beetle bodies (Table 1). This is not surprising, considering that CTD is typically concentrated in meloid reproductive organs [28].

Our bioassays demonstrated strong effects from *B. majalis* extracts and CTD against nearly all species tested. Population growth of the parasitic protozoan *Trichomonas vaginalis* was strongly suppressed, with 50% growth inhibition (GI_50_) at 75.7 (body extract), 15.5 (hemolymph), and 5.6 µg/mL (CTD), with consistent dose-responses (Table 2). CTD had a remarkable activity level in comparison with other natural products and extracts screened against this parasite [45].

Previously, CTD showed promising effects against *Leishmania major*, both in vitro, with 80% growth inhibition at a concentration of 50 µg/mL, and in vivo in experimentally infected BALB/c mice [46,47]. In addition, norcantharidin and analogs displayed good antiplasmodial activity on sensitive (D6) and chloroquine resistant (W2) strains of *Plasmodium falciparum*, with IC_50_ values close to 3.0 µM [48]. However, ours is the first report on the effects of *B. majalis* extracts and cantharidin on *Trichomonas* sp.

The plant endoparasitic nematode *Meloidogyne javanica* was also very sensitive to the hemolymph extract, with CTD being extremely potent against this parasite (CTD showed 26 and >1000 times more potent LD_50_ and LD_90_ values, respectively, than hemolymph) (Table 3). The activity of the hemolymph did not correlate with its content in CTD, may be due to the lipophilic nature of the extract. This is the first report on the effects of *B. majalis* hemolymph and CTD on nematodes, and specifically on *M. javanica*. Preliminary results on the larvicidal effect of CTD analogs on the parasitic nematode *Haemonchus contortus* lead to the proposal of serine/threonine phosphatase inhibitors as potential nematicidal targets [49,50]. 

*B. majalis* extracts and cantharidin were strong antifeedants against aphids, with *Rhopalosiphum padi* more sensitive than *Myzus persicae*. The antifeedant effects correlated with the CTD content of the extracts (hemolymph > body extract), with pure CTD the strongest aphid antifeedant (Table 4). The feeding behavior of the polyphagous chewing lepidopteran *Spodoptera littoralis* was not affected (data not shown). Previously, cantharidin showed toxicity and growth-regulation effects against *Plutella xylostella* moth caterpillars [51], inhibited glutathione S-transferase from Codling moth caterpillars, *Cydia pomonella* [52] and lepidopteran protein phosphatases [53,54]. Furthermore, CTD and several acylthiourea derivatives showed contact toxicity against the aphid *Brevicoryne brassicae* [55]. However, this is the first report on the antifeedant effects of CTD and CTD-rich extracts against aphids.

The extracts and CTD were effective ixodicidal agents with similar effective doses (Table 5). CTD and CTD-rich extracts had LD_50_ concentrations seven times more potent than the positive control, nootkatone [56] but had similar LD_90_ values. 

The fact that the hemolymph and the body extracts showed similar effects suggests the presence of additional ixodicidal components in the body extract, which had the lowest CTD concentration. Chemically defended insects often contain multiple classes of toxic compounds [44]. Wang et al. (2014) [55] previously reported the acaricidal effects of CTD on *Tetranychus cinnabarinus*. This is the first report on the ixodicidal effects of cantharidin.

## 3. Conclusions

Our study documents that CTD is active against a diverse range of organisms including protozoa, nematodes, ticks, and insects. As such, our findings support Bravo et al.’s (2014) [36] hypothesis that Great Bustards might reduce parasite loads via the ingestion of blister beetles, a possible example of self-medication.

On a broader scale, our results (and those of other authors) showing CTD activity against a diverse range of taxa, suggest that this natural product might be developed to combat specific pests or pathogens under certain conditions. Of course, CTD is highly toxic to humans and many vertebrates [57,58], and appropriate safety precautions must be followed. That CTD is currently used in humans to treat common and molluscum warts, to remove tattoos, and as a counterirritant, and that it has commonly been used to encourage livestock breeding, implies that additional (safe) uses might be found.

## 4. Materials and Methods

### 4.1. Insect Extracts

Two hundred adult *Berberomeloe majalis* of mixed sex (males averaged 22.3 ± 0.3 mm and 490 ± 34.1 mg; females 30.3 ± 1.4 mm and 1234 ± 138.3 mg) were collected in Central Spain (Finca La Garganta, Ciudad Real) in March 2015. Insects were frozen at −20 °C until use. To obtain hemolymph, we cut the terminal abdomens and allowed hemolymph to drip into a vial. The resulting hemolymph (4.8 mL) was extracted with dichloromethane (DCM) (10 mL, 3 times) and the solvent evaporated to give an extract of beetle hemolymph (20 mg). The remaining insect bodies (~172 g of combined males and females) were macerated with DCM (400 mL, 3 times) at room temperature, filtered, and the solvent evaporated to give 44 g of body extract.

### 4.2. Cantharidin Quantification

We analyzed the above extracts via GC–MS (Thermo Finnigan Trace GC 2000 coupled with a Trace MS mass selective detector). The chromatographic conditions were controlled using Xcalibur software version 1.2 (Thermo Finnigan, San Jose, CA, USA). The GC column was a SLB-5 ms (30 m, 60.32 mm, 0.25 µm, Supelco Analytical, Bellefonte, PA, USA). The flow rate of helium was 0.8 mL/min. The injection volume was 1 mL in splitless mode for 2 min. Injector conditions were 250 °C in constant flow mode. The column oven had an initial temperature of 50 °C for two minutes. The subsequent temperature was programmed at a heating rate of 10 °C/min to 310 °C. The final temperature was held isothermally for 5 min. Total run time was 30 min. Cantharidin (CTD) detection was performed by selected ion monitoring (SIM), registering m/z = 128 (= the majority ion of CTD´s mass spectra). CTD identification was performed by comparison with mass spectra available in the NIST MS search 2.0 library. The extracts were dissolved in DCM and the concentrations injected were 0.1 and 0.5 mg of extract/mL DCM for the hemolymph and body extracts, with an injection volume of 1 mL. Confirmation and quantification were achieved with the retention time and calibration curves (range: 0.015–48 mg/mL, slope: 476.401, r^2^ = 0.999) obtained from the injection of CTD standard purchased from Sigma–Aldrich (St. Louis, MO, USA).

### 4.3. Bioassays

In all bioassays we tested various doses of three primary preparations: (1) Extract of beetle hemolymph. (2) Extract of beetle bodies. (3) 99% pure CTD (Sigma–Aldrich, St. Louis, MO, USA).

### 4.4. Antiprotozoal Activity

Antiprotozoal activity was evaluated on the metronidazole-sensitive *Trichomonas vaginalis* JH31A no.4 isolate (American Type Culture Collection, (ATCC)). The flagellates were cultured in a trypticase-yeast extract-maltose modified medium supplemented with 10% heat-inactivated fetal bovine serum (FBS) and antibiotic solutions at 37 °C and 5% CO_2_. Assays were carried out in glass tubes containing 10^5^ trophozoites/mL. After 5–6 h of seeding, extracts (1), (2) or CTD were added to log-phase growth cultures at several concentrations (500, 250, 100, 50, 25 and 10 μg/mL for extracts (1) and (2); 100, 50, 25, 10, 5 and 1 μg/mL for CTD). The tubes were incubated for 24 h at 37 °C and 5% CO_2_. The trichomonacidal activity was obtained by a fluorimetric method using resazurin (Sigma-Aldrich) as previously described [59,60]. The experiments were performed at least two times in triplicate. GI_50_ values, as well as the 95% CI were calculated by Probit analysis (SPSS v.20, IBM, Armonk, NY, USA).

### 4.5. Nematicidal Activity

We tested the toxicity of our three preparations against 2nd stage juveniles (J2) of *Meloidogyne javanica* previously maintained on *Lycopersicon esculentum* plants (var. “Marmande”) in pot cultures at 25 ± 1 °C and >70% RH. The experiments were carried out in 96-well microplates (Becton, Dickinson), as described per Andrés et al., (2012) [61]. The three preparations were tested at initial concentrations of 1.0 and 0.5 mg/mL (final concentration in the well) and diluted serially if necessary. The number of dead juveniles was recorded after 72 h. All treatments were replicated four times. The data were determined as mortality percentage corrected according to Scheider–Orelli’s formula. Effective lethal doses (LC_50_ and LC_90_) were calculated for the active pure compounds by Probit analysis (five serial dilutions, 0.5–0.01 mg/mL).

### 4.6. Insect Bioassays

*Spodoptera littoralis*, *Myzus persicae* and *Rhopalosiphum padi* colonies were reared on an artificial diet, bell pepper (*Capsicum annuum*) and barley (*Hordeum vulgare*) respectively, and maintained at 22 ± 1 °C, >70% relative humidity with a photoperiod of 16:8 h (L:D) in a growth chamber. Choice antifeedant bioassays were conducted in Petri dishes with newly emerged *S. littoralis* sixth instar larvae (at least 10 replicates with 2 insects each to give a SE < 10) or 2 × 2 cm plastic boxes with adults (24–48 h old) of the aphids *M. persicae* and *R. padi* (20 replicates with 10 insects each). Feeding or settling inhibition on treated (10 µL of extract or CTD solution) and untreated (10 µL of solvent) leaf disks of the host plant (%FI or %SI) were calculated as %FI= (1−(T/C)) × 100, where T and C are the consumption of treated and control leaf disks, respectively, or as %SI= (1−10 (%T/%C)) × 100 where %C and %T are percent aphids settled on control and treated leaf disks, respectively, as described [62]. The antifeedant effects (%FI/%SI) were analyzed for significance by the non-parametric Wilcoxon signed-rank test. EC_50_ (effective dose to obtain 50% feeding inhibition) were determined for extracts and CTD (four serial dilutions, 10–1 or 5–0.5 mg/mL) with %FI/%SI values > 75% from a linear regression analysis (%FI/%SI values on log (Dose)) (statistical package: www.statgraphics.com).

### 4.7. Ixodicidal Activity

*Hyalomma lusitanicum* engorged female ticks were collected in central Spain (Finca La Garganta, Ciudad Real) from their host (deer) and maintained at 22–24 °C and 70% RH in a growth chamber until oviposition and egg hatch. Resulting larvae (4–6 weeks old) were used for the bioassays. Briefly, 50 µL test solution were added to 25 mg of powdered cellulose at different concentrations and the solvent evaporated. For each test, three replicates with 20 larvae each were used. Larval mortality was checked after 24 h of contact with the treated cellulose in the environmental conditions described [56], using a binocular magnifying glass. The mortality data shown have been corrected with respect to the control according to Schneider–Orelli’s formula. Effective lethal doses (LC_50_ and LC_90_) were calculated by Probit Analysis (5 serial dilutions, STATGRAPHICS Centurion XVI, version 16.1.02). 

## Figures and Tables

**Figure 1 toxins-11-00234-f001:**
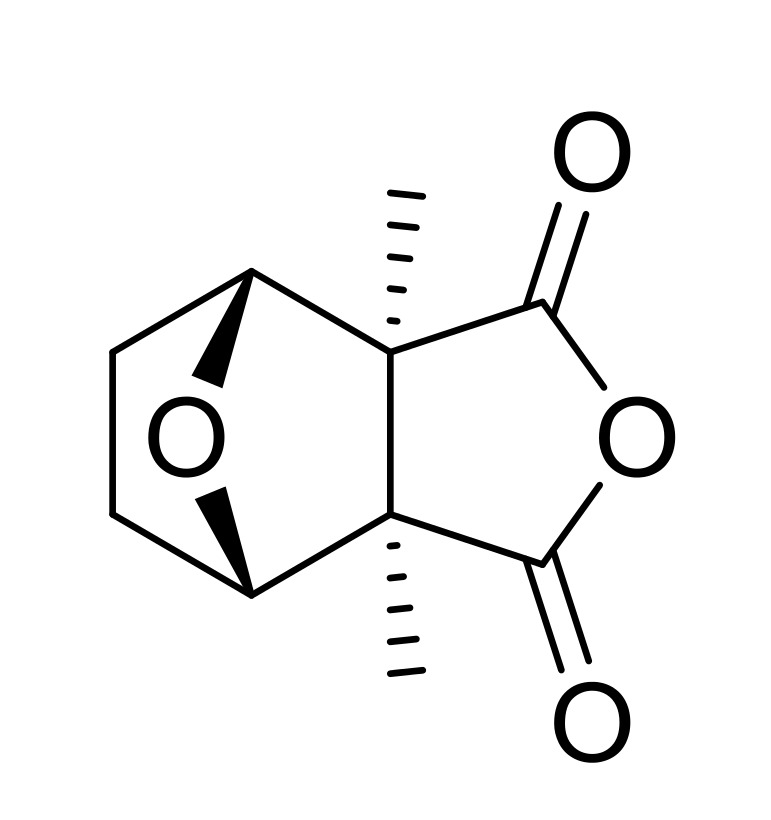
Cantharidin.

**Figure 2 toxins-11-00234-f002:**
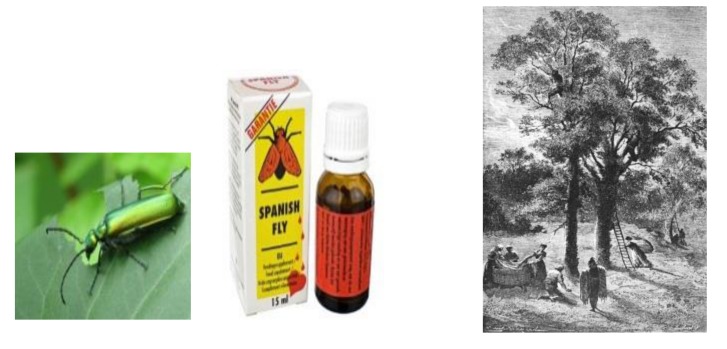
Spanish fly (*Lytta vesicatoria*) (from Stefanie Hamm), an example of commercial cantharadin preparation, and collecting blister beetles in Spain in the 17th Century.

**Table 1 toxins-11-00234-t001:** Cantharidin (CTD) concentration in *Berberomeloe majalis* blister beetles and their extracts.

Extract	CTD (µg/mg)	Total CTD (mg) ^a^	Distribution of Total CTD (%)	CTD Per Beetle (mg) ^b^
Body	41.2	1813	99.3	9.06
Hemolymph	295.0	5.9	0.7	0.03
Total	-	1819	100	9.1

^a^ Total CTD (for 20 mg and 44 g of hemolymph and body extract respectively). ^b^ Estimated CTD per beetle, N = 200 beetles.

**Table 2 toxins-11-00234-t002:** Activity of cantharidin (CTD) and extracts of *Berberomeloe majalis* blister beetles against the parasitic flagellated protozoan, *Trichomonas vaginalis*. Data are expressed as percentages of growth inhibition.

Concentration (µg/mL)	Body Extract	Hemolymph Extract	CTD	Metronidazole
500	92.7 ± 0.8	99.8 ± 0.3	-	-
100	52.7 ± 4.6	77.5 ± 4.1	98.1 ± 0.3	-
GI_50_ (µg/mL) (95% CL)	75.7 (24.6–220.2)	15.5 (1.4–36.2)	5.6 (4.2–7.0)	0.6 (0.3–1.4)

**Table 3 toxins-11-00234-t003:** The effects of cantharidin (CTD) and *B. majalis* extracts on juvenile mortality in the parasitic nematode *Meloidogyne javanica.*

Treatment	Dose (µg/µL)	Mortality ^a^ %	Lethal Concentrations ^b^	
			LC_50_ (µg/mg)	LC_90_ (µg/mg)
**Body**	**1**	74.9 ± 2.92	nc	
Hemolymph	1	84.05 ± 2.64	0.656 (0.626–0.687)	1.108 (1.054–1.172)
CTD	0.5	100 ± 0	0.0252(0.023–0.027)	0.065 (0.061–0.070)

^a^ Data corrected according to Scheider–Orelli’s formula. Values are means of four replicates. ^b^ Lethal doses to give 50% and 90% mortality (95% Confidence Limits).

**Table 4 toxins-11-00234-t004:** Insect antifeedant effects of *B. majalis* extracts and cantharidin (CTD).

Treatment	Concentration (µg/cm^2^)	*Rhopalosiphum Padi*	*Myzus Persicae*
		%SI ^b^	
**Body**	**50**	94.35 ± 2.42	82.75 ± 8.28
	EC_50_ ^b^	6.7 (4.63–9.63)	14.3 (8.1–25.3)
Hemolymph	50	96.84 ± 1.94	93.23 ± 5.0
	EC_50_ ^b^	0.8 (0.5–1.5)	3.38 (1.98–5.77)
CTD	50	94.7 ± 3.5	91.50 ± 2.31
	EC_50_ ^b^	0.098 (0.031–0.3)	0.211 (0.05–0.91)

^a^ %SI = (1 − (T/C)) × 100, where T and C are settling on treated and control leaf disks. ^b^ EC_50_, effective dose to give a 50% inhibition (95% Confidence Limits).

**Table 5 toxins-11-00234-t005:** Effects of *B. majalis* extracts and cantharidin (CTD) on *Hyalomma lusitanicum* tick larval mortality.

Treatment	Mortality ^a^	Lethal Concentrations ^b^	
		LC_50_ (µg/mg)	LC_90_ (µg/mg)
**Body**	81.7 ± 0.9	12.79 (10.84–14.93)	23.93 (20.79–28.94)
Hemolymph	70.2 ± 1.8	12.25 (10.65–13.94)	21.05 (18.74–24.47)
CTD	90 ± 0.1	12.84 (11.55–14.30)	20.31 (18.32–23.11)
Nootkatone ^c^	-	4.02 (1.92–7.42)	18.02 (13.60–29.16)

^a^ At 20 µg/mg cellulose. Data corrected according to Scheider–Orelli’s formula. Values are means of three replicates. ^b^ Lethal doses to give 50% and 90% mortality (95% Confidence Limits). ^c^ From Ruiz–Vázquez et al. [56].

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
