# Peer review of "Antiparasitic Properties of Cantharidin and the Blister Beetle *Berberomeloe majalis* (Coleoptera: Meloidae)"

_toxins, 2019, doi:10.3390/toxins11040234_

Round 1

Reviewer 1 Report

This article provides interesting information on the parasitic activity of cantharidin obtained from the blister beetles. The authors hypothesized that beetle consumption by the great bustards reduces the parasite load of the bird. For this, they tested a bunch of different parasites including different micro-organisms and insects. 1) what was the rationale behind using these organisms for study? do they all infest the great bustards? I do not see this study providing a piece of direct evidence for beetle consumption as antiparasitic in the birds. 2) Were any studies done with treating the bird with cantharidin or feeding the blister beetle and reduction in parasite load was observed? 

I would suggest adding more information on the biology of the blister beetles. Since the authors describe extracting this toxin from the hemolymph and leftover bodies, the information regarding the source of this chemical within the beetles and how this chemical is produced would be more interesting for the readers.  

Methods have not been described in details and seem many steps are missing. 

Abstract needs to be improved as well. 

Further suggestions are enclosed in the pdf attached.  

Reviewer 2 Report

Comments are in the .pdf file: toxins-489090-peer-review-v1 Dont' Panic.pdf
